# Effect of Aging Treatment on Microstructural Evolution and Mechanical Properties of the Electron Beam Cold Hearth Melting Ti-6Al-4V Alloy

**DOI:** 10.3390/ma15207122

**Published:** 2022-10-13

**Authors:** Jiaxin Yu, Zhengpei Yin, Zhirong Huang, Shuai Zhao, Haiguang Huang, Kun Yu, Rongfeng Zhou, Han Xiao

**Affiliations:** 1Faculty of Materials Science and Engineering, Kunming University of Science and Technology, Kunming 650093, China; 2National-Local Joint Engineering Laboratory for Technology of Advanced Metallic Solidification Forming and Equipment, Kunming University of Science and Technology, Kunming 650093, China; 3Yunnan Titanium Industry Co., Ltd., Chuxiong 651209, China

**Keywords:** TC4 alloy, EBCHM, aging treatment, microstructure, mechanical properties

## Abstract

Ti-6Al-4V (Ti64 or TC4) alloy is widely used in the industrial field. However, there have been few studies of the TC4 alloy melted by electron beam cold hearth melting (EBCHM) technology. Aging treatment has a considerable influence on the secondary α-phase in titanium alloys. Therefore, TC4 alloy melted by EBCHM technology was investigated in this study. The effect of different aging times on the microstructural evolution and mechanical properties of titanium alloy sheets was evaluated. The results showed that, with increase in aging time, the primary α-phase enlarged and grain globularization occurred. In addition, some transformed β-phases disappeared. The strength and Vickers hardness of the heat-treated sheets decreased, while the plasticity increased with increase in aging time, indicating that the mechanical properties developed with evolution of the microstructure. After aging at 560 °C for 2 h, the properties overall were optimal. The type of fracture of the samples was ductile fracture; the dimples became larger with increase in aging time. After heat treatment, the recrystallized nucleus, substructures and HAGBs increased, while the deformed structure and LAGBs decreased. Some grains had rotated following heat treatment, indicating that anisotropy was greatly reduced.

## 1. Introduction

Ti-6Al-4V (Ti64 or TC4) alloy is widely used in the aerospace and chemical industries, in ships and other fields due to its low density, high specific strength, good corrosion resistance and biocompatibility [1,2,3]. TC4 titanium alloy is a representative α + β titanium alloy, which has α- and β-phases [4]. It is one of the most used titanium alloy materials [5].

Traditional titanium alloy production mostly obtains plates by forging, machining and rolling following a triple-vacuum arc remelting (VAR) process [6]. However, this smelting method produces many inclusions. VAR melting is a slow process as the liquid flow condition is unsuitable for acceleration of the dissolution of inclusions. High-density inclusions sink to the bottom of the pool, where the temperature is lower than at other positions. Compared to the triple VAR melting process, electron beam cold hearth melting (EBCHM) technology has significant advantages for the production of very clean metal. During the melting process, volatile impurities evaporate due to exposure in a vacuum atmosphere (10^−2^~10^−3^ Pa). The melt-flow conditions lead to a long inclusion residence time, which promotes the dissolution of inclusions [7]. If EBCHM technology is used, the content of inclusions can be effectively reduced [8]. Moreover, EBCHM technology has the advantages of high energy density and high vacuum, which can improve the purity of titanium alloys [9]. The ingots obtained by EBCHM technology can be rolled directly after melting, which simplifies the production process and reduces the cost. Sheets rolled from TC4 alloy ingots melted by EBCHM technology can be used in aviation, weapon manufacture and other fields. Many studies have shown that stress can be reduced and the microstructure of the alloy can be improved by appropriate heat treatment [10,11]. The microstructure plays an important role in the mechanical properties of the material, and can be adjusted to improve mechanical properties of the material [12,13,14]. The strength and hardness of titanium alloys can be greatly improved using a supersaturated solid solution formed by solution treatment [15,16]. The dispersed phase can be precipitated by subsequent aging treatment to improve strengthening, and the plasticity and toughness can be improved at the same time [17,18]. The microstructure of titanium alloys is greatly affected by the process [19,20]. To enable titanium alloys to be better used in industrial production, many investigations have been carried out into the structure and properties of titanium alloys. Fan et al. [21] investigated the effect of solution aging at 950 °C/AC/1 h + 540 °C/AC/4 h on the microstructure and properties of a Ti-6Al-4V alloy produced by selective laser melting. The results showed that a basket-weave structure can be obtained after solution with aging treatment, resulting in excellent mechanical properties. Lei et al. [22] investigated the effects of different structures on the tensile properties and impact toughness of Ti-6Al-4V alloy. The results showed that a lamellar structure had higher impact toughness and strength, but plasticity was lower than for a globular structure. Chen et al. [23] studied the effect of aging heat treatment on the microstructure and tensile properties of a high strength titanium alloy. The results showed that the β grain size can be restrained by a primary α-phase during aging treatment, and the morphology of the secondary α is sensitive to time. Xu et al. [24] studied the microstructural evolution and mechanical properties of a titanium alloy during solution-plus-aging treatment. The results showed that an α+β solution-plus-aging treatment resulted in an excellent combination of strength and plasticity.

The TC4 alloy is a very widely used alloy. Hence, it is important to investigate for TC4 alloy low-cost production technology. If EBCHM technology is used, it can not only reduce the content of inclusions, but can also provide technical methods enabling low-cost production. To date few studies have investigated TC4 alloy ingots melted by EBCHM technology, so, results are not available for direct industrial application. Aging treatment has a significant influence on the secondary α-phase precipitated in titanium alloys, which results in their different properties. Therefore, it is crucial to study the microstructure evolution of titanium alloys because of its potential applications. In this study, TC4 alloy melted by EBCHM technology was used and the effect of aging treatment for different times on the microstructure and mechanical properties of TC4 alloy sheets was investigated. This provided not only a theoretical basis for the production of TC4 alloy sheets with excellent performance and low cost, but also a protocol for the subsequent heat treatment process.

## 2. Experimental Procedures

The experimental material was TC4 alloy. A billet was cut in the middle part of the ingot with a size of 80 × 70 × 50 mm^3^, which was produced by an electron beam cold hearth melting process at an average melting rate of 500 kg/h. A melting power of 200~210 kW was used to melt the alloy, and a power of 180~185 kW was used to ensure a constant temperature in the surface. The vacuum pressure was 0.1~1.0 Pa during the melting process. The dimension of the crystallizer was 1270 × 220 × 200 mm^3^ and the size of the cold hearth was 1450 × 400 × 250 mm^3^. The hot-rolled sheet was obtained by first rolling along the length direction and then rolling along the width direction. The chemical composition of the titanium alloy sheet was measured by chemical analysis, as shown in Table 1. The β-phase transformation temperature was obtained by differential scanning calorimetry (DSC) of the alloy at about 958.6 °C, as shown in Figure 1. The samples were further heat-treated with solution-plus-aging treatment. First, the sheet was heated to 930 °C and kept for 15 min, then cooled in water for solution treatment, kept at 560 °C for 2 h, 3 h and 4 h, and then furnace cooled (FC) for aging treatment. The standard tensile specimens built in a horizontal plane for tensile property evaluation, as shown in Figure 2, were cut in the rolling direction (RD) and transverse direction (TD).

A small part of the geometry of 10 × 10 × 10 mm was used for microstructure analysis and electron backscattered diffraction (EBSD) tests. Standard Kroll’s reagent (HF: HNO_3_: H_2_O = 3:5:12) was used to identify the various phases. The microstructure and fracture morphology of specimens were observed using an ECLIPSE MA200 microscope and a ZEISS EVO18 (Zeiss, Oberkochen, Germany) scanning electron microscopy (SEM). Image-Pro-Plus software was used to calculate the phase volume fraction. X-ray diffraction was used to measure the Cu target Kα radiation; the scanning rate was 10°/min. Tensile tests were performed using a Zwick/Roll-Z150 machine (Zwick/Roll, Ulm, Germany); the tensile rate was 1.8 mm/min. Each group of specimens was repeated three times and the average value taken. Vickers hardness tests were performed using an HMV-G21S machine (Shimadzu, Kyoto, Japan) with a load of 200 g and a loading time of 15 s; each group of specimens was repeated seven times and the average value taken.

## 3. Results and Discussion

### 3.1. Microstructural Characterization of TC4 Aged Sheets

The purpose of aging treatment is to decompose the metastable phase formed after heat treatment. Solution treatment and then quenching in water to effect aging is performed to precipitate the secondary phase with a dispersed distribution to enable strengthening [15,17,18]. The microstructures of the TC4 alloy which were obtained by solution treatment and aging treatment for different times (2 h, 3 h, and 4 h) are shown in Figure 3. It can be seen from the figure that the structure was composed of a large number of α-phase colonies and a small number of β-phase colonies, with a small amount of recrystallized equiaxed α-phase dispersed in it. From Figure 3a, b, it can be seen that the thick α-phase colonies in the longitudinal section and the cross-section of the hot-rolled sheet were distributed, and that the β-phase was distributed among them with the same degree of distribution. The average grain sizes of the equiaxed α-phase in the longitudinal section and cross-section were about 1.52 μm and 1.33 μm, respectively. The microstructures after aging for 2 h in the longitudinal section and cross-section are shown in Figure 3d, e. The microstructure consisted of a few equiaxed α-phases and some β phases. The average grain sizes of the equiaxed α-phase in the longitudinal section and cross-section were about 2.32 μm and 2.21 μm, respectively. For the transverse cross-section, the SEM images of the alloy are shown in Figure 4. It can be seen from Figure 4b that many secondary α-phases were precipitated on the transformed β matrix; the volume fraction of the primary equiaxed α-phase was about 31.2%. As shown in Figure 3g,h, the average grain sizes of the equiaxed α-phase in the longitudinal section and cross-section were about 2.83 μm and 2.69 μm, respectively. It can be seen from Figure 3g,h that the microstructure was composed of many primary α-phases (equiaxed and lamellar) and transformed β phases, and thick α laths were distributed along the rolling direction. It can be seen from Figure 4c that the microstructure was composed of many primary equiaxed α-phases and transformed β phases; the volume fraction of the primary equiaxed α-phase was about 31.0%. Compared with the microstructure after aging for 2 h, the primary α phase, lamellar α-phase and secondary α-phase on the β matrix had merged and developed. As shown in Figure 3j,k, the average grain sizes of the equiaxed α-phase in the longitudinal section and cross-section were about 2.98 μm and 2.77 μm, respectively. It can be seen from Figure 3j,k that, with prolongation of the aging time, the microstructure was still composed of some equiaxed primary α-phases and a few transformed β phases. The volume fraction of the equiaxed α-phase was calculated according to the area fraction of the phase in the figure. It can be seen from Figure 4d,e that the average grain size of the equiaxed primary α-phase increased. This phenomenon reflected the grain growth of the α phase; however, the volume fraction of the primary α-phase did not change too much and its value was about 30.2%. The α-phase tended to be equiaxed, indicating that grain globularization had occurred, and the content of the transformed β-phase decreased gradually. To sum up, the structural evolution of the sheet under different aging times is shown in Figure 5. With increase in aging time, the average grain size of the primary equiaxed α-phase increased gradually and grain globularization occurred; however, the volume fraction of the primary α-phase was not obviously changed. The morphology of the transformed β-phase was obviously changed with increase in aging time; the secondary α-phase precipitated on the β matrix appeared to be merged and matured, and the content of the transformed β-phase gradually decreased. This phenomenon was also observed in other titanium alloys [24,25].

Figure 6 shows the energy spectrum of the sheet obtained after aging for 2 h. It can be seen from Figure 6b that the element Ti in the hot-rolled sheet after solution-plus-aging treatment was uniformly distributed in the structure, and there was no segregation. As shown in Figure 6c,d, the elements Al and V were also uniformly distributed in the structure, and there was basically no aggregation of Al or V elements at the grain boundaries of the equiaxed α phase.

The XRD patterns of the sheets are shown in Figure 7a. It can be seen from the figure that the phase compositions of the hot-rolled and heat-treated sheets were mainly α/α′-Ti and β-Ti [26] and exhibited the highest diffraction peak at 40.5°. The peak intensities of each phase of the heat-treated sheet were significantly higher than that of the unheated sheet. With increase in aging time, the peak intensities of (
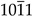
) increased gradually, indicating that the volume fraction of α/α′-Ti increased after solution-plus-aging treatment. As can be seen from the structural evolution, the structure became coarse and matured with increase in aging time, so the intensities of each peak were enhanced to different degrees. In order to investigate the effect of the aging time on the grain size, the Scheler equation was used to calculate the crystallite size of (
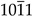
) [27]:(1)$$D=\frac{K\lambda }{\beta_{hkl}cos\;\theta }$$
where *D* is the crystallite size (nm), *K* is a constant (0.89 is used in this case), *λ* is 0.154 nm, *β_hkl_* is the FWHM of diffraction peak, *θ* is the peak position, and all angles are in radians. It can be seen from Figure 7b that the FWHM decreased, and the calculated crystallite size perpendicular to (
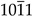
) was increased with increase in aging time. The results showed that the crystallite size of (
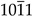
) increased with increased aging time, showing the same trend as for grain size.

### 3.2. Mechanical Properties of TC4 Aged Sheets

The TC4 alloy sheets obtained by heat treatment were subjected to tensile tests unidirectionally at room temperature in the RD and TD directions. The mechanical properties are shown in Figure 8. It can be seen from the figure that the tensile strength (UTS), yield strength (YS) and elongation (EL) of the hot-rolled sheets in the RD and TD directions were 912.7 and 941.5 MPa, 792.1 and 884.3 MPa, and 11.2% and 9.4%, respectively. The UTS, YS and EL of the sheets after aging for 2 h were 1073.6 and 1077.4 MPa, 955.0 and 991.0 MPa, and 10.4% and 10.5%, respectively. After aging for 3 h, they were 1038.1 and 1048.7 MPa, 922.3 and 962.4 MPa, and 10.5% and 10.7%, respectively. They were 1036.2 and 1041.8 MPa, 917.0 and 920.0 MPa, and 10.7% and 11.3% after aging for 4 h, respectively. With increase in aging time, the strength of the sheet decreased and the plasticity slightly increased This corresponded to the evolution of the microstructure [15,17]. The differences in UTS, YS and EL of the hot-rolled and heat-treated sheets in RD and TD directions were: 28.8 MPa, 92.2 MPa and 1.8% (hot-rolled); 3.8 MPa, 36.0 MPa and 0.1% (2 h); 10.6 MPa, 40.1 MPa and 0.2% (3 h); 5.6 MPa, 3.0 MPa and 0.6% (4 h). The difference between the anisotropy of the sheets obtained after different aging times was small, and the comprehensive properties of the sheet after aging for 2 h were the best. The effect on the anisotropy of the sheets was similar to other titanium alloys [13,28]. Compared with the hot-rolled sheet, the UTS and YS of the sheet obtained after aging for 2 h in the RD and TD directions increased by 17.63% and 14.43%, 20.57% and 12.07%, respectively. The EL decreased by 7.14% in the RD direction and increased by 11.70% in the TD direction. In summary, as the aging time increased, the strength of the heat-treated sheets decreased and the plasticity increased slightly. After aging for 2 h, the sheet had the best comprehensive properties, the UTS and YS of the sheet increased, and the EL decreased compared with the hot-rolled sheet.

The Vickers hardness values for the TC4 alloy sheets are shown in Figure 9. It can be seen that the Vickers hardness value decreased gradually with increase in aging time. The sheet obtained after aging for 2 h had the largest Vickers hardness which was 308.8 HV. The reason for the decrease in Vickers hardness may be that, as the aging time increased, its structure and secondary α-phase on the β matrix had merged and grown up, and the transformed β had basically disappeared [3,29]. As a result, its strength and hardness were reduced.

The fracture morphologies of the samples are shown in Figure 10. The macroscopic fracture morphologies in the RD and TD directions were relatively uneven and showed a dark gray fiber shape, which had a certain degree of necking. The fractures of the hot-rolled and heat-treated samples were composed of dimples with different sizes; they were deep and there were many small dimples in the large dimples. Moreover, the fracture dimples became larger with increase in aging time and all the samples were ductile fractures. The small dimples were formed by the secondary α, flaky α phase, and β phase, and the large dimples were formed by the equiaxed α-phase [17]. With increasing aging time, the volume fraction of the secondary α-phase and transformed β-phase declined, and the volume fraction of the equiaxed α-phase increased. Therefore, the fracture dimple became larger with increase in aging time. This phenomenon was also observed in other studies [3,4]. In summary, the fracture dimples were ductile fractures in both the RD and TD directions and the plasticity was improved due to the dimples becoming larger with increase in aging time.

### 3.3. Evolution of Grain and Orientation Distribution after Heat Treatment

To explore the recrystallization, grain boundaries and texture changes after heat treatment, the hot-rolled sheet and the sample obtained after aging for 2 h were characterized by EBSD testing. The internal average misorientation angle (IAMA) map and the proportion of hot-rolled sample are shown in Figure 11a,c; the figures for the heat treatment sample are shown in Figure 11b,d. In the IAMA maps, the angles which are above 1° are classified as deformed grain (red), the angles which are under 1° and the angles from subgrain to subgrain which are above 1° are classified as substructure grains (yellow). The remaining grains are classified as recrystallized grains (blue) [30]. It can be seen from the figure that, after solution-plus-aging treatment, the recrystallized nucleus and substructures were significantly increased, and the deformed matrix structure was significantly reduced. The deformed matrix structure of the sheet was reduced from 91% to 18.24%; the recrystallized nuclei and substructure increased from 7.57% to 50.63% and 1.43% to 31.13%, respectively. The increase in recrystallization nuclei was due to recrystallization during the heat treatment process. The formation mechanism may be that, during the heat treatment process, a certain section of the original grain boundary penetrated the larger residual strain grains and the deformation storage energy disappeared [10,29]. Moreover, many strain-free recrystallized nuclei were formed through the merger of subgrain or grain boundaries [31,32].

Figure 12 shows the misorientation angle distributions of TC4 alloy sheet after heat treatment. The misorientation distribution map represents the proportion of orientation angles of grain boundaries; the random distribution of polycrystalline orientation is represented by the black line, indicating that it has no preferred orientation. The grain boundaries significantly changed after heat treatment, the low-angle grain boundaries (LAGBs, <5°) and high-angle grain boundaries (HAGBs, >15°) of the hot-rolled sheet accounted for 90.19% and 9.81%, and the LAGBs and HAGBs of the heat-treated sheet accounted for 38.83% and 61.17%, respectively. Moreover, a large number of LAGBs in the hot-rolled sheet were distributed in the range 0° to 15°, showing a characteristic “single peak”, and there were some obvious peaks in the heat-treated sheet around 4°, 64° and 90°. There were a number of grain boundary orientations around 60°, which was related to the different variants of the β-phase to the α phase. The α-phase precipitated in a Burgers relationship to the β-phase with twelve crystallographic variants; the misorientations could be divided into six types according to the axis-angle pairs, as shown in Table 2 [33]. The distribution of the sheet after heat treatment was similar to random distribution, but there was a certain deviation. Compared with the hot-rolled sheet, the HAGBs of the heat-treated sheet increased and the degree of preferred orientation decreased. In summary, the HAGBs increased and the LAGBs decreased after solution-plus-aging treatment. The reason was that recrystallization occurred during the heat treatment process, which caused the original lamellar structure to be spheroidized [25,34]. In addition, the HAGBs of the sheet after solution-plus-aging treatment accounted for the largest proportion, indicating that its degree of recrystallization was the largest; the degree of preferred orientation was reduced, showing the characteristic of random orientation.

The pole figures (PFs) are shown in Figure 13. The hot-rolled sheet had a strong basal texture in the {0001} pole figure, and the c-axis of the grains was deflected away from the ND direction to the RD direction; the angle between the c-axis and ND direction was about 40°. There was variant distribution and coarsening behavior of the α-phase [33]. In this study, the intensity of the heat-treated sheet was lower than that of the hot-rolled sheet, and the c-axis of the α-phase grains was almost parallel to the TD, which showed typical transverse texture characteristics [34]. In summary, the hot-rolled and heat-treated sheets both had basal texture in the {0001} pole figure, and the intensity was obviously weakened after heat treatment. In addition, the grains rotated and the transverse texture appeared [35].

The inverse pole figures (IPFs) are shown in Figure 14. It can be seen from Figure 14a that the grain orientation in the RD direction of the hot-rolled sheet was between <0001> and 
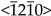
, and the grain orientations in the TD and ND direction were between <0001> and 
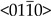
. It can be seen from Figure 14b that most of the grains in the RD direction of the heat-treated sheet were between 
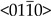
 and 
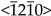
, and some grains were between <0001> and 
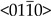
 in the TD direction. The grains in the ND direction were oriented away from <0001>, and their intensity decreased after heat treatment, indicating that some grains had rotated. This phenomenon represents the weakening and randomization of the texture [36].

## 4. Conclusions

(1)After solution-plus-aging treatment, the phase of the TC4 hot-rolled and aged sheets was mainly composed of α/α′-Ti and β-Ti, and the peak intensities of each phase were higher than those of the unheated sheet, indicating that the grains had grown up. With increase in aging time, the equiaxed primary α-phase grew up gradually and grain globularization occurred, but the volume fraction of the primary α-phase did not change significantly. The morphology of the transformed β-phase was obviously changed; many secondary α-phases precipitated on the β matrix appeared to be merged and grown up, and some transformed β-phases disappeared.(2)With increase in aging time, the strength of the heat-treated sheets decreased and the plasticity slightly increased; the difference between the anisotropy of the sheets was small. After aging for 2 h, the comprehensive properties were the best. The Vickers hardness of the sheet showed a downward trend; the value was proportional to its strength, and inversely proportional to its elongation. The fracture dimples, which were ductile fracture, in the RD and TD directions of the samples were similar, and the dimples became larger with increase in aging time.(3)After heat treatment, the recrystallized nucleus and substructures were increased, and the deformed matrix structure was reduced. The LAGBs decreased and the HAGBs increased after heat treatment. There was a basal texture in the {0001} pole figure, the intensity was obviously weakened after heat treatment, and some grains had rotated, indicating that the anisotropy was greatly improved.

## Figures and Tables

**Figure 1 materials-15-07122-f001:**
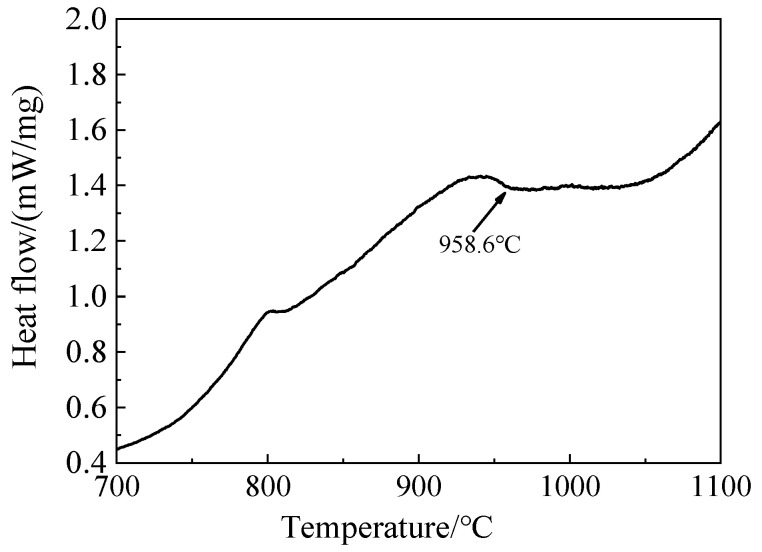
DSC curves of TC4 alloy.

**Figure 2 materials-15-07122-f002:**
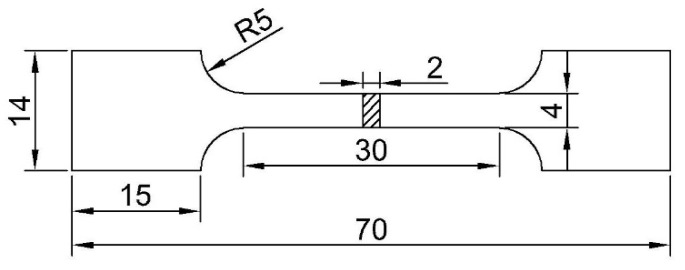
The size of tensile specimen (mm).

**Figure 3 materials-15-07122-f003:**
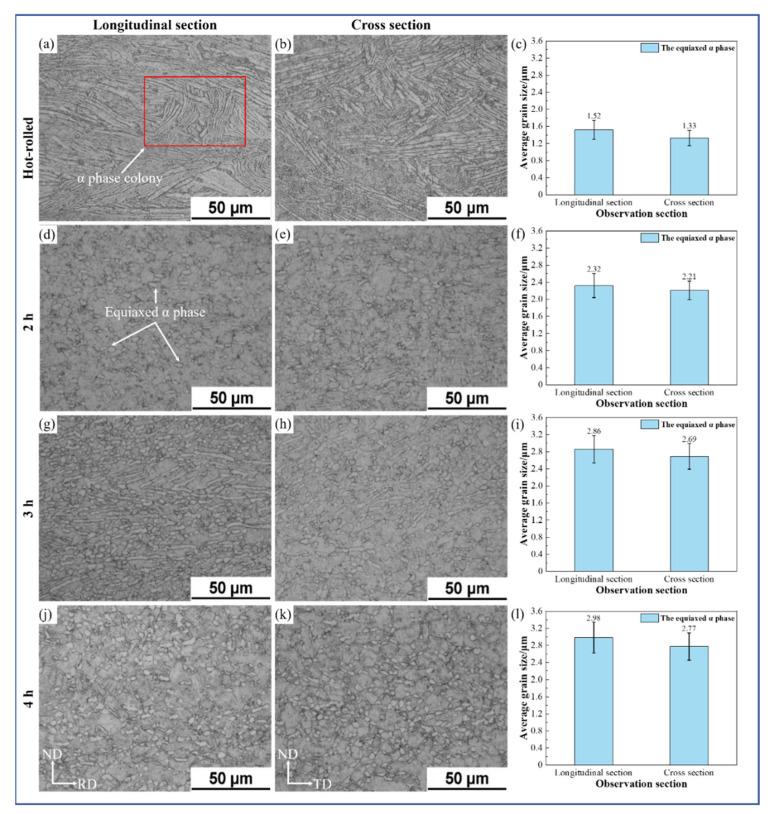
Microstructure and average grain size of TC4 alloy aged sheets.

**Figure 4 materials-15-07122-f004:**
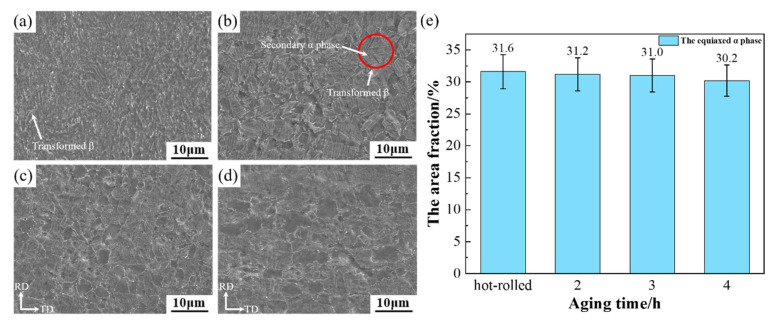
SEM images of TC4 alloy aged sheets: (**a**) Hot-rolled (**b**) 2 h (**c**) 3 h (**d**) 4 h (**e**) the area fraction of equiaxed α phase.

**Figure 5 materials-15-07122-f005:**
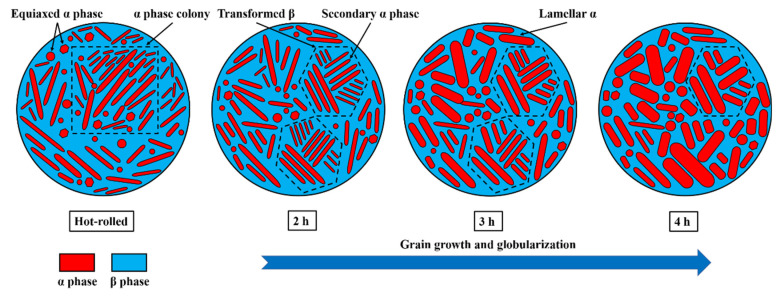
Microstructure evolution of TC4 alloy aged sheets.

**Figure 6 materials-15-07122-f006:**
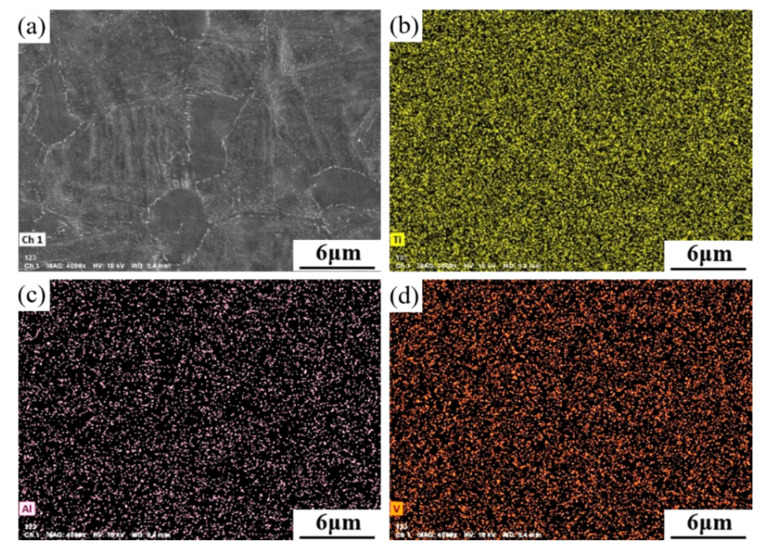
EDS of the sheet aging for 2 h: (**a**) Scanning surface (**b**) Ti (**c**) Al (**d**) V.

**Figure 7 materials-15-07122-f007:**
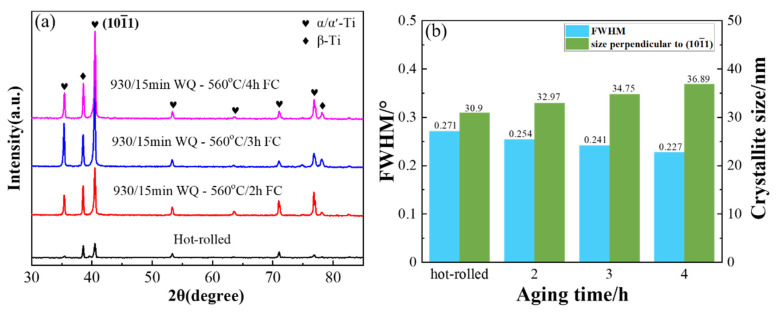
(**a**) XRD patterns of TC4 alloy aged sheets (**b**) the FWHM of (
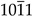
) and the calculated size.

**Figure 8 materials-15-07122-f008:**
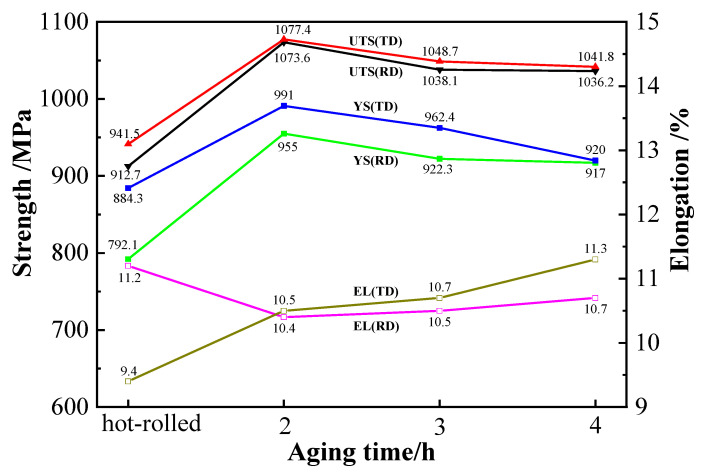
The mechanical properties of TC4 alloy aged sheets.

**Figure 9 materials-15-07122-f009:**
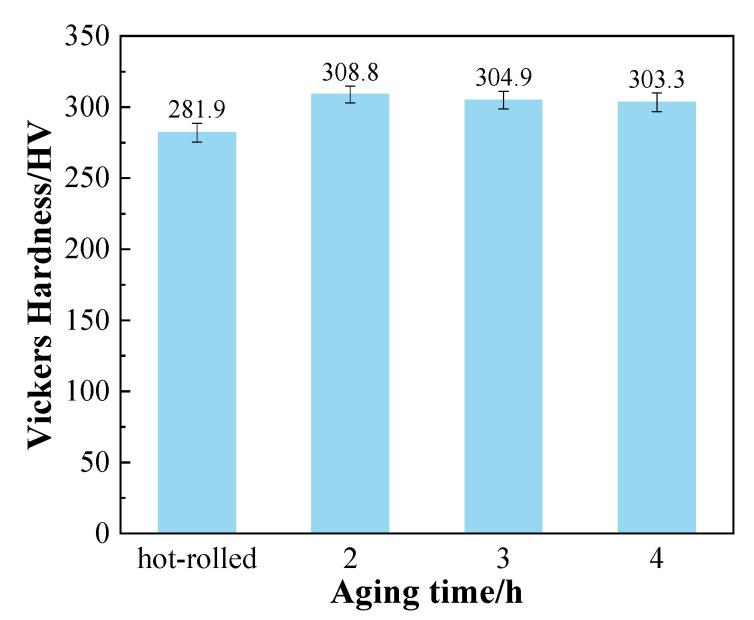
Vickers hardness of TC4 alloy aged sheets.

**Figure 10 materials-15-07122-f010:**
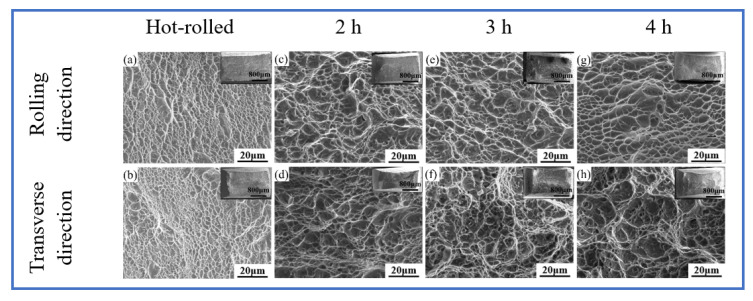
Fracture morphologies of TC4 alloy aged sheets.

**Figure 11 materials-15-07122-f011:**
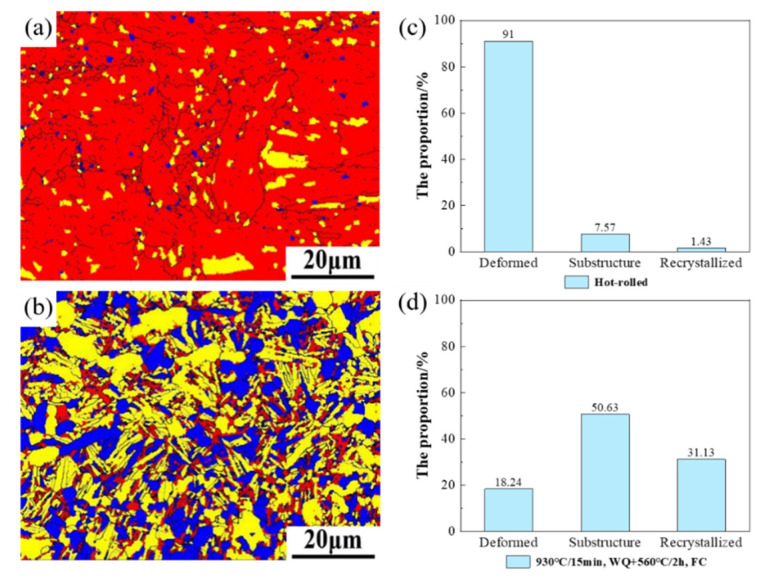
Recrystallized (blue), substructure (yellow), and deformed (red) grains in TC4 alloy: (**a**,**c**) Hot-rolled; (**b**,**d**) 930 °C/15 min, WQ + 560 °C/2 h, FC.

**Figure 12 materials-15-07122-f012:**
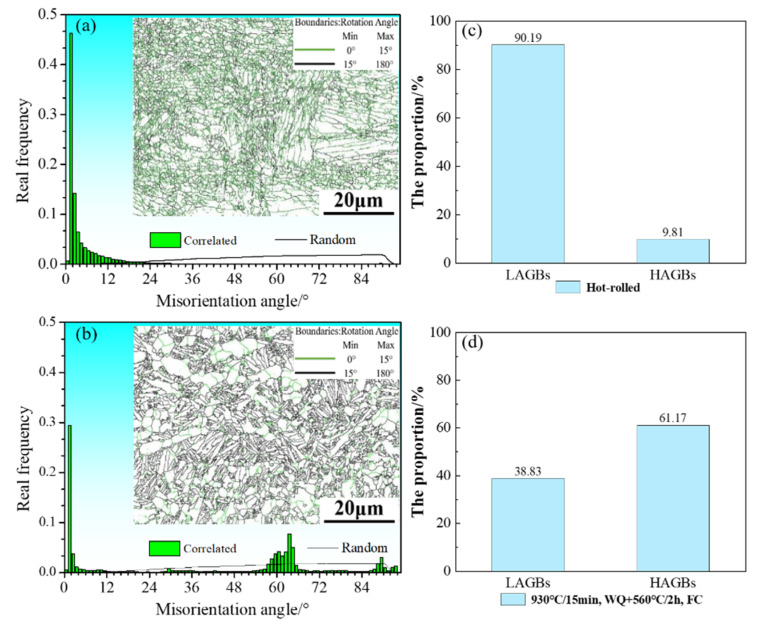
Misorientation angle distributions of TC4 alloy sheets: (**a**,**c**) hot-rolled; (**b**,**d**) 930 °C/15 min, WQ + 560 °C/2 h, FC.

**Figure 13 materials-15-07122-f013:**
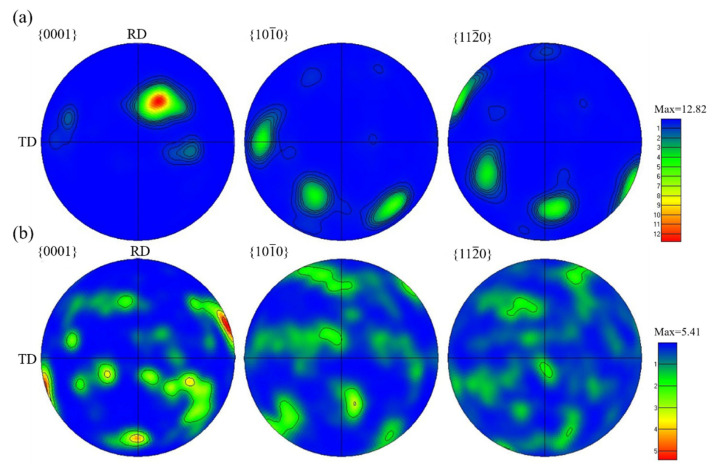
PFs of TC4 alloy sheets: (**a**) Hot-rolled (**b**) 930 °C/15 min, WQ + 560 °C/2 h, FC.

**Figure 14 materials-15-07122-f014:**
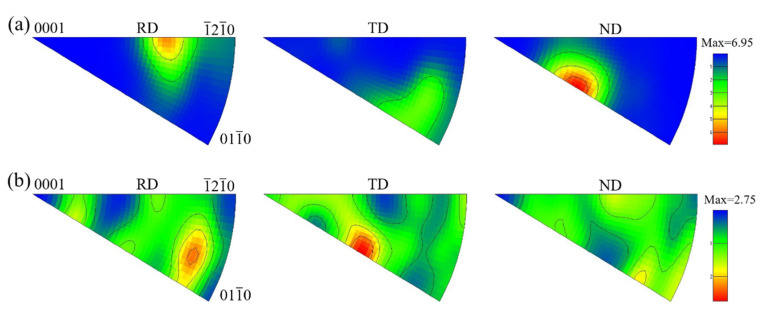
IPFs of TC4 alloy sheets: (**a**) Hot-rolled (**b**) 930 °C/15 min, WQ + 560 °C/2 h, FC.

**Table 1 materials-15-07122-t001:** Chemical composition of TC4 alloy (wt, %).

Ti	Al	V	Fe	C	N	H	O
Bal.	5.93	4.27	0.07	0.05	0.009	0.004	0.14

**Table 2 materials-15-07122-t002:** The six types of pair-wise misorientation between α and β phase.

Type	Angle Pair	Axis	Probability Occurrence
A	0		15.38%
B	10.53°	[0 0 0 1]	7.69%
C	60°	[1 1 −2 0]	15.38%
D	60.83°	[−1.377 −1 2.377 0.359]	30.77%
E	63.26°	[−10 5 5 3]	15.38%
F	90°	[1 −2.38 1.38 0]	15.38%

## Data Availability

The data presented in this study are available upon reasonable request from the corresponding.

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
