# Peer review of "Effect of Aging Treatment on Microstructural Evolution and Mechanical Properties of the Electron Beam Cold Hearth Melting Ti-6Al-4V Alloy"

_materials, 2022, doi:10.3390/ma15207122_

Round 1

Reviewer 1 Report

This manuscript describes the effect of heat treatment on microstructure and mechanical properties of TC4 titanium alloy after Electron Beam Cold Hearth Melting (EBCHM) treatment. The authors have employed a variety of tools such XRD, EBSD, SEM, Tensile tool to analyse TC4 alloys before and after heat treatment. Solution aging treatment of TC4 (or Ti64) alloys have been carried out extensively in the literature. Although EBCHM treatment is less studied, one similar work has been carried out in 2008 (“An Investigation on Electron Beam Cold Hearth Melting of Ti64 Alloy”, https://doi.org/10.1016/S1875-5372(10)60004-5).

The authors need justify the novelty of their work and consider the following comments, before this manuscript can be accepted for publication.

1.       The paper cited above is a good reference. You need justify what is your novelty over that to justify the value of your manuscript.

2.       What is the melting rate (in kg/hr) you used for EBCHM in this work?

3.       TC4 or Ti64 typically comprises of primary alpha HCP phase and secondary beta BCC phase. What do you mean by secondary alpha phase in the text? You need define it clearly.

4.        Line 80-81: what kind of “chemical analysis” method do you use to derive composition in Table 1? Pls specify.

5.       Fig. 12: no explanation about what Figure (c & d) represents for.

6.       Fig. 4: no explanation what (e) is for

7.       Fig. 6: you may want to standardize “3H” to “3h”. Same for 2H and 4H.

8.       The English throughout the manuscript needs be improved significantly. There are many simple typos throughout the manuscript. A few examples are given below:

a.       Line 19-20: “were occurred, besides, some 19 transformed β phases disappeared.” à “were occurred. Besides, some 19 transformed β phases disappeared.”

b.       Line 33-34: “which has the α and β phases [4],. mMoreover, it is one of the most commonly used titanium alloy materials [5].

c.       Line 36-37: “However, this smelting method will produce many inclusions;. Iif the electron beam cold hearth melting (EBCHM) technology is used, the content of inclusions can be effectively reduced

d.       Line 46-47: “[14,15],. and Tthen the dispersed phase can be”

e.       Line 52-54: Fan et al. [20] does not study Ti-6Al-V but Ti6Al-4V.

Reviewer 2 Report

In this work, the authors studied the effect of aging treatment on microstructural evolution and mechanical properties of a TC4 alloy elaborated by Electron Beam Cold Hearth Melting. For this purpose, different heat treatment conditions have been chosen and different characterization techniques including XRD, EBSD have been used to investigate the microstructural evolution. Attention is also paid to the relation between the mechanical properties and the associated microstructure. The subject is interesting and the paper could be considered for publication after its quality is improved.

Some general advice is as follows:

-        While explaining some of the results, it would be better if the authors could (a little) further explain the underlying mechanisms/reasons

-        The paper should be fully checked to improve the overall quality

In addition, the reviewer has the following detailed questions/remarks:

-        Check the list of authors, remove “and” at the end of the list

-        Line 19 (and somewhere): “…the grain globularization were occurred”, remove “were”

-        Line 25: As HAGB and LAGB are used for the first time in the form of abbreviation, their full names should be given first. The authors should check other similar cases (for example SLM, EBSD) and correct them.

-        Line 26: “…the anisotropy was greatly improved” is not clear. It is better to write “… the anisotropy was greatly reduced”.

-        Line 54: Is it “Ti-6Al-V” or “Ti-6Al-4V”?

-        Line 79: What does “after length and width reversal” mean? It should be clearer. Further explanation could be given.

-        Line 94: What does “…small type of part geometries” mean?

-        Line 94: Use “analysis” instead of “analyzation”

-        Line 96 and Line 97: It should be “The microstructure and the fracture morphology…were observed using…”

-        Line 99: Check the sentence “The characteristic of X-ray diffraction was using…”

-        Line 111: It should be “…which were obtained by….”

-        Line 124: it should be “In the transverse cross section, the SEM…”

-        Line 162: “With the increase of aging time, the intensities of each peak increased gradually, indicating that the grains grew up…”, could you please explain why?

-        Line 173: What does “crystallite size of (10-11)” mean?

-        Figure 4b: Indicate The area fraction of “what”, because this information is neither in the figure nor in the caption, and we have to look at the corresponding text to know.

-        Figure 6: In the text, it seems that Fig. 6 is cited before Fig. 5. It is better to change the order of these two figures.

-        Line 187: What does “drawn” mean?

-        Lines 230 and 231: Check the sentence

-        Line 254: “…had significantly changed…”

-        Figure 12: There is a number of grain boundaries of around 60°. They correspond to twins? It is better to give more explanation/interpretation about this phenomenon.

-         Line 228: “…the fracture dimple became larger with the increase of aging time…”. If further explanation can be given about the underlying reasons, it would be better.

Round 2

Reviewer 1 Report

I thank the authors for making improvement to the manuscript, which has improved the manuscript. The manuscript still needs consider some comments below, before it can be accepted for publication:

1.       Although the authors added in the reference (https://doi.org/10.1016/S1875-5372(10)60004-5) I recommended as Ref 7, the authors failed to explain what is the novelty of the work compared to this paper published 14 years ago.

2.       The calculated melting rate of TC4 titanium alloy (about 400~600 kg/h) should be added to the manuscript for readers’ information. The previous paper’s melting rate of TC4 titanium alloy is 70, 100 and 140 kg/h. Your calculated melting rate is much higher than this paper’s one. How does that affect the Al volatilization losses of the EBCHR melted ingots?

3.       In the Abstract and first line of Introduction: “Ti-6Al-4V (TC4)” can be changed to “Ti-6Al-4V (Ti64 or TC4)”

4.       In the “Keywords” section, you can use “TC4” to replace “Ti-6Al-4V” as you have used TC4 throughout the manuscript

5.       Fig. 4(e) is not being mentioned at all in the main manuscript, although it is in the figure. And how is the “the area fraction of equiaxed α phase” being determined in Fig. 4(e)? It should be explained in the manuscript.

6.       Fig. 11: It is good to label what (a) and (b) represents for, instead of asking readers to figure them out from Fig. (c) and (d). Meanwhile, how is the “deformed”, “recrystallized” and “substructure” being determined here, as LAGB and HAGB are only explained in Fig. 12? Why is the so high “deformed” grain after hot rolling?

7.       What is the angle definition between LAGB and HAGB?

8.       Fig. 12 (c & d): should X-axis use LAGB instead of LAB and HAGB instead of HAB?

9.       Line 306: “Tab.2” should be spelled out in full name.

Reviewer 2 Report

Thanks for having revised the paper. The revised version is much improved and it can be considered for publication.

Author Response

Thank you very for your affirmation and constructive suggestion.

Round 3

Reviewer 1 Report

I thank the authors for the detailed revision. The readability has been improved substantially. As this manuscript focuses on EBCHM technology, it is good for it to provide more information about the EBCHM technology, such as the total electron beam power (kW), the diameter of the solidifying crucible and the dimension of the cold hearth, etc.
